The movement and distribution of pregnant spotted ragged-tooth sharks, Carcharias taurus, in the iSimangaliso Wetland Park, South Africa

Cerqueira Sara C. 1 saritacerqueira@live.com.pt
http://orcid.org/0000-0001-8357-5157 Olbers Jennifer Margaret 2 3
Smith Grant 4
Carpenter Michelle 4
Pereira Mário J. 1
http://orcid.org/0000-0003-1790-6055 Cliff Geremy 2 5
1 Department of Biology & CESAM, Universidade de Aveiro , Aveiro , Portugal
2 Wildtrust , Pietermaritzburg, KwaZulu-Natal , South Africa
3 Nelson Mandela University , Gqeberha, Eastern Cape , South Africa
4 Sharklife Conservation Group , Sodwana Bay, KwaZulu-Natal , South Africa
5 School of Life Sciences, University of KwaZulu-Natal , Durban , South Africa
Yapıcı Sercan
Electronic publication date: 2024 Dec 20
Publication date: 2024
Volume: 12
Electronic Location ID: e18736
Received 2024 Sep 25; Accepted 2024 Nov 28
Copyright: © 2024 Cerqueira et al.
Copyright year: 2024
Copyright holder: Cerqueira et al.
License: This is an open access article distributed under the terms of the Creative Commons Attribution License, which permits unrestricted use, distribution, reproduction and adaptation in any medium and for any purpose provided that it is properly attributed. For attribution, the original author(s), title, publication source (PeerJ) and either DOI or URL of the article must be cited.
License URL: https://creativecommons.org/licenses/by/4.0/

Keywords: Aggregation site, Database, Gestation, Movements, Photo identification, iSimangaliso Wetland Park, Marine Protected Area, Reproduction cycle, Population assessment

Funding: Grant Smith of Sharklife Conservation Group Grant Smith of Sharklife Conservation Group was the primary funder for data collection. The funders had no role in study design, data collection and analysis, decision to publish, or preparation of the manuscript.

==============================
The spotted ragged-tooth shark, Carcharias taurus, is widely distributed in subtropical continental coastal seas. In South Africa, it is commonly found along the entire south and east coasts, including the iSimangaliso Wetland Park (IWP) in the far north, which is the largest Marine Protected Area on the South African coast. Pregnant females occur there for much of the year, with the largest aggregations in summer. It is here we used remote underwater photography (RUP), supplemented with in-situ surveys to photo-identify individuals, using unique spot patterns. Three known aggregation sites (Raggie Reef, Quarter-Mile Reef and Mushroom Rocks) were monitored over a 5-year period between 2018 and 2023. We photo-identified 574 individuals (569 females and five males) and registered 1,200 sightings, using images of the right flank. The identification of new individuals persisted throughout the study, with the discovery curve showing no signs of reaching an asymptote. A total of 97% (n = 550) of females observed were noticeably pregnant. Individuals were consistently identified across all sample years and at all three reefs, exhibiting movements among the three monitored sites. The reproductive cycle is generally regarded as 2 years, but some females appeared to have a 2-year rest between pregnancies. Raggie Reef, which lies in the sanctuary zone, emerged as the reef with the highest index of popularity, as individuals were present almost constantly (90% of the sampling days). The findings of this study confirm the crucial role that the IWP plays in the conservation of a species that is globally Critically Endangered.

Introduction

The spotted ragged-tooth shark, Carcharias taurus Rafinesque, 1810, inhabits subtropical and tropical coastal waters off continental land masses in the Atlantic, Pacific and Indian Oceans, as well as warm-temperate waters of the Mediterranean Sea (Smith & Pollard, 1999; Compagno, 2001; Ebert, Dando & Fowler, 2021). It favours inshore rocky reefs, where it occurs close to the seabed, often in caves and under overhangs (Pollard, Lincoln-Smith & Smith, 1996). This species reproduces by oophagous viviparity, and exhibits intrauterine cannibalism, giving birth to two embryos (one per uterus) after a gestation of 9 months, followed by a year of recovery (Bass, 1975; Gilmore, Dodrill & Linley, 1983). This biennial reproductive cycle, coupled with a very small litter size, leads to low reproductive output, making it vulnerable to anthropogenic impacts, particularly overfishing. In addition to fishing threats, the species faces habitat loss and degradation in inshore coastal waters, as well as the possible impacts of climate change (Rigby et al., 2021). As a result, the species was globally classified as Critically Endangered on the IUCN Red List in 2020 (Rigby et al., 2021). In South Africa, the commercial sale of C. taurus is prohibited, making it illegal to sell any part of the species (Government of South Africa, 1998). A population estimate from mark-recapture data over a decade ago suggested a South African adult population of C. taurus at 16,700 individuals (Coefficient of Variation = 9%) (Dicken, Booth & Smale, 2008). Klein et al. (2020) indicated that this population remains healthy, potentially representing the last stable subpopulation of C. taurus globally.

Aggregation behavior is common among elasmobranchs, serving various functions such as facilitating courtship (Whitney, Pratt & Carrier, 2004), reducing predation risk (Guttridge et al., 2012), enhancing foraging efficiency (Dewar et al., 2008) and providing reproductive benefits (Wearmouth et al., 2012). Similar to other shark species, C. taurus demonstrate philopatry to certain areas (Hueter et al., 2005). In Australia, aggregations have been recorded at numerous specific sites, but there is only one documented site where females congregate during their pregnancy from September to January (Bansemer & Bennett, 2009). In the southwest Atlantic, pregnant individuals aggregate in subtropical waters of Brazil (Sadowsky, 1970), and after parturition, they move south where they rest, in cooler waters (Lucifora, Menni & Escalante, 2002).

The South African population of C. taurus is regularly found at specific sites alongside the east and south coasts, from False Bay in the extreme south to northern KwaZulu-Natal (KZN), and very occasionally on the west coast (Bass, 1975; Smale, 2002). The KZN Sharks Board’s catch and tagging data have been used to ascertain the distribution and migratory routes of this species in South Africa (Wallett, 1974; Dudley, 2002). Adults of both sexes move northwards from the Eastern Cape into southern and central waters of KZN, where mating takes place in late October and November (Dicken, Smale & Booth, 2006a; Olbers & Cliff, 2017). Adult females in this population are under a well-defined reproductive migration. After mating, females continue to swim north to spend their pregnancy in northern KZN and southern Mozambique (Bass, 1975; Smale, 2002), most likely to take advantage of the seasonally warm water which accelerates metabolism and development (Bass, 1975; Bansemer & Bennett, 2009; Lucifora, Menni & Escalante, 2002). The Agulhas Current, in the western Indian Ocean, transports warm tropical waters southwards along the east and south coasts of South Africa (Schumann, 1998). This is responsible for the warm water temperatures in the iSimangaliso Wetland Park (IWP), a Marine Protected Area, which provides an optimal environment for gestating C. taurus. Then, the near-term pregnant individuals return south, around July and August, to the Eastern Cape (Wallett, 1974), where parturition takes place in specific nursery locations from September to November (Smale, 2002; Dicken, Smale & Booth, 2006a).

To date, the known aggregation sites of pregnant females in South Africa have not been closely monitored, highlighting a gap in our understanding of the life history of this species. Here, we compiled a dataset of individuals using photo-identification of individuals in the IWP over a 5-year period. We identified the gender of these individuals and visually assessed the status of pregnancy among females. We used interannual resightings to ascertain the duration of the species’ reproductive cycle. We monitored local movement and distribution at three different reefs, to determine their carrying capacity and popularity. Monitoring critical habitats for the reproduction of a vulnerable species, like C. taurus, is essential to develop target conservation efforts. This study highlights the importance of specific sites for gestation, providing insights that can support the conservation of these crucial sites and clarify the role of the IWP as a key habitat for their reproductive activities.

Materials and Methods

Study areas

Based on the Hoschke & Whisson (2016) definition where aggregation sites for C. taurus are locations where five or more sharks gather frequently each year, this study focusses on three known aggregation sites in the IWP: Raggie Reef (RR), Quarter-Mile Reef (QM) and Mushroom Rocks (MR) located on Seven-Mile Reef (Fig. 1).

Figure 1 Map depicting the study sites, in a Marine Protected Area.

Map of South Africa highlighting KwaZulu-Natal province and the specific locations of the three study sites, within the iSimangaliso Wetland Park (IWP). The study sites include Raggie Reef (RR), Quarter-Mile Reef (QM) and Seven-Mile Reef (7M), with a focus on the Mushroom Rocks (MR) area.

QM is located close to the public launch site (<1 km), situated between 500–800 m offshore at a depth of 10–12 m. RR is approximately 43 km south of the public launch site, and in comparison to QM, it is a far larger reef, approximately 250 × 180 m, at a depth of 10–14 m. Seven-Mile Reef is approximately 11 km to the north of the launch site, 800 m offshore, at a depth of 14–25 m. Even though it is bigger than QM and RR, being 1,400 × 390 m, only a small portion of this reef was monitored, designated MR—as this is the most common congregating site on Seven-Mile Reef. QM and MR are in the Sodwana Diving Restricted Zone (SDRZ), a protected area in the Sodwana Bay region. In contrast, RR is situated in the iSimangaliso Offshore Wilderness Zone (IOWZ), a sanctuary where no angling, scuba diving or spearfishing are permitted.

The iSimangaliso Wetland Park Authority and Ezemvelo KZN Wildlife approved the field work with an authorized Research Agreement. The Department Of Forestry Fisheries and Environment authorized the research (Permit RES2023/56).

Sampling

Sampling was carried out over five consecutive seasons from 2018/2019 to 2022/2023 from September to March, coinciding with the peak abundance of C. taurus. The first sampling season, which began in December 2018, was an exception. In this study, summer months were considered as December, January and February.

Remote underwater photography

GoPro Hero3 or Hero4 cameras were used for the remote underwater photography (RUP) (Rezzolla, Boldrocchi & Storai, 2014). Each photographic unit (Fig. 2) consisted of two cameras, positioned at 90° to one another to broaden the field of view and set to take images in wide angle at 30-s intervals. Each camera was connected to an external battery (50 Ah or 100 Ah), through a USB cable. The two cameras and batteries were placed inside a sealed Plexiglass tube housing (20 cm in diameter and 25 cm in height). The camera housing was mounted 1 m above the seabed on a vertical pole which was secured to a 45 kg base. Freedivers deployed units and retrieved them for servicing and cleaning every 6–12 days. Each pair of cameras was manually synchronized to capture images at the same time.

Figure 2 Components and deployment of the remote underwater photographic (RUP) system.

(A) RUP housing equipped with GoPro cameras and batteries, specifically designed for underwater imaging. (B) Unit without underwater housing. (C) The RUP system installed at the study site, actively capturing images.

The RUPs were deployed at each site for different sampling durations. The RUP at QM was first deployed in 2019/2020 season, and every sampling season thereafter. At RR, the RUP was first deployed in 2020/2021 and for the subsequent two sampling seasons, while the RUP was only installed at MR, during the fourth sampling season, 2021/2022.

Diver observations—in-situ surveys

During in-situ surveys, freedivers used handheld GoPro Hero5 or Hero6 cameras to capture images of C. taurus whenever visibility and circumstances allowed. They also counted the maximum number (MaxN) of sharks present at the reefs. Weather and sea conditions were a very important factor in dive observations. QM was surveyed almost every day during the summer months, since it was the most accessible reef. It was first surveyed in the first sampling season, 2018/2019. Surveys at RR were undertaken every 10 days, with the longest gap being 12 days. This reef was first surveyed during the second sampling season, 2019/2020. As MR was a deeper reef, scuba surveys were conducted instead of snorkel surveys in 2019/2020 and in 2021/2022.

Photo-identification

All the images were downloaded, and only the ones suitable for photo-identification were selected. The criteria for selection were that the shark’s entire right flank, as well as both dorsal fins and the pelvic fin present in the frame. Only the images of the right-side were used, to avoid double counting (Meekan et al., 2006; Van Tienhoven et al., 2007; Holmberg, Norman & Arzoumanian, 2008; MacKey et al., 2008). The images were then analyzed with the I3S Classic version 4.02 software as described by Van Tienhoven et al. (2007) (Interactive Individual Identification System, https://reijns.com/i3s/) to identify each individual, based on their natural spot patterns—requiring a minimum of 12 identifiers and a maximum of 30. This necessitated selecting three reference points on the body in each image: the base of the first dorsal fin, the base of the second dorsal fin and the base of the pelvic fin (Fig. 3).

Figure 3 Reference and identification points for photo-identification of Carcharias taurus.

The three blue points are used by the I3S as reference points for the photo identification process: the first dorsal fin near the top, the second dorsal fin midway along the back, and the pelvic fin on the underside. The red points indicate distinctive spots on the shark’s body, which are unique to each individual.

The darkest spots were chosen first, as they were most likely to be seen in all conditions. After selecting all key points, the image of the individual was compared against those already in the database. Each comparison produced a score, with lower scores indicating a closer positive match to a known individual. Higher scores typically suggested different individuals, although this could vary depending on factors like lighting and image quality. Conversely, lower scores were generally more reliable in indicating matches, with very low scores often reflecting a high likelihood of the same individual. Photo-identification methods can lead to two potential errors: (1) false positive, incorrectly matching two unique individuals as the same one or; (2) false negative, where a known individual is not identified, thus classifying it as a new one. Therefore, a visual comparison was always necessary, to compare specific spots on the two sharks, and the individual was either matched positively or recorded as a new individual. To minimize errors of subjectivity, the combination of the I3S automated system and a manual pattern–matching exercise was undertaken by the same observer. After the identification was complete, the images of all females were examined, in order to detect any evidence of pregnancy. If the female had mating scars and/or a noticeably distended abdomen, with lateral bulges, it was considered to be pregnant (Bansemer & Bennett, 2009). While this assessment can be considered as subjective, these visual indicators are commonly used to infer pregnancy in C. taurus and are generally reliable in natural settings (Bansemer & Bennett, 2009). Furthermore, there is considerable unpublished evidence suggesting that these aggregation sites are likely used by females primarily for gestation.

Index of popularity and density of each site

The index of popularity reflects the relative frequency of shark occurrence at the aggregation site, irrespective of the number of sharks. In order to determine the density or carrying capacity of each site, based on the maximum observed number of sharks (MaxN), we only used the sharks counted during in-situ surveys. This was necessary because RUP imagery has severe limitations and only provides a field of view of up to 180° from a fixed point.

Results

Sampling effort

The total sampling effort combines data from both RUP and in-situ surveys, representing the total number of days where observations were undertaken per reef, as well as per season (Table 1). QM experienced the highest effort (n = 484 days), followed by RR (n = 359 days) and lastly, MR (n = 55). Sampling effort per month was uneven, with the summer months (December, January and February) being the most heavily sampled between 17 to 31 days.

Table 1 Sampling effort (number of sampling days), for both remote underwater photography (RUP) and in-situ surveys.

This table presents the sampling effort across study sites over the 5-year study period. It includes the number of days each site was sampled per sampling season, along with the total sampling effort for each site.

		Monitoring period (s)	Total sampling effort	
Sampling effort (number of observation days)	Reefs	18/19	19/20	20/21	21/22	22/23		
RR	–	9	100	142	108	359	
QM	69	108	132	86	89	484	
MR	–	23	–	32	–	55	
Note:

Raggie Reef (RR), Quarter-Mile Reef (QM), Mushroom Rocks (MR).

Identification of individuals with photo-identification

In this study, 574 individuals were identified between 2018 and 2023 (Fig. 4). The discovery curve shown has not yet reached an asymptote, with a steady increase in the number of newly identified individuals over the 5-year study. There were 1,200 shark sightings of 574 identified individuals. It is important to note that a sighting is recorded each time an individual shark is identified, even if it has been identified before. For example, if an individual shark is identified on one day and then identified again on the next day, this counts as two sightings for that one individual. Most (62.9%, n = 358) individuals were only sighted once, but 37.1% (n = 213) were resighted at least once, predominantly in the same sampling season (Fig. 5), with a single individual sighted 19 times across the 5 year-long study period. Of the resighted individuals, only 2.1% (n = 12) were resighted in a subsequent sampling season. The sex ratio of the population was 99.0% (n = 569) females and 1% (n = 5) males.

Figure 4 Cumulative discovery curve of newly identified Carcharias taurus individuals.

The cumulative discovery curve depicts the number of newly identified individuals of Carcharias taurus in the iSimangaliso Wetland Park, per sampling season, from 2018 to 2023.

Figure 5 Total number of newly identified individuals of Carcharias taurus, alongside the number of individuals resighted during that season and respective ratio.

Pregnancies

During the 5-year survey period, 97.0% of females were classified as pregnant, based on mating scars and/or a distended belly (n = 550), while the status of 3.0% was uncertain (n = 19). In some pregnant individuals who were seen several times in a particular season, it was possible to see the belly enlarging with time (Fig. 6). Of the 550 pregnant females identified over the entire survey period, 12 were resighted in different sampling seasons, and were always considered to be pregnant (Table 2). Three females were resighted with a 1-year interval between sightings and nine females were resighted with a 2-year interval.

Figure 6 Carcharias taurus: pregnancy observation.

(A) Female individual seen at RR on 21st November 2021, with no visible signs of pregnancy, and (B) the same individual, seen at RR on 24th March 2022, with clear signs of pregnancy.

Table 2 Pregnancy history of resighted Carcharias taurus’ females.

Reproductive patterns of females, including the season of their first pregnancy recorded in this study, years in between sightings, and second pregnancies observed, over the 5-year sampling period.

ID	Monitoring period (s)	
18/19	19/20	20/21	21/22	22/23	
27_Ct_F	1st	*	2nd	–	
37_Ct_F	1st	*	2nd	–	
49_Ct_F	1st	*	2nd	–	
99_Ct_F	–	1st	Resting years	2nd	
106_Ct_F	–	1st	*	2nd	
122_Ct_F	–	1st	*	2nd	
281_Ct_F	–	1st	*	2nd	–	
334_Ct_F	–	1st	*	2nd	
375_Ct_F	–	1st	*	2nd	
403_Ct_F	–	1st	*	2nd	
489_Ct_F	–	–	1st	*	2nd	
511_Ct_F	–	–	1st	*	2nd	
Note:

1st, first pregnancy observed; *, years without sighting such individual; 2nd, second pregnancy observed.

Description of the sharks’ movements within the IWP

Over the entire study period, 29 of the 213 individuals that were resighted were found at a different reef from where they were first detected. There was a total of 33 movements, of which 19 were in a northerly and 14 in a southerly direction. There are no significant differences between northward and southward movements (χ2: 2, n = 33, p = 0.34). Of the northward movements, 11 were from RR to QM, three from QM to MR and five from RR to MR. Of the 14 southward movements, there were 11 from QM to RR, two from MR to QM and one from MR to RR. The majority of northward movements (n = 15) were between December and January, while the southerly movements were mostly between February and March (n = 9).

Index of popularity and shark abundance of each site

Throughout the survey period, C. taurus were observed on 223 out of 484 (46.1%) sampling days at QM. In contrast, sharks were recorded on 325 out of 359 sampling days at RR, constituting a 90.5% presence. Insufficient sampling was conducted at MR to determine an index of popularity.

Sightings of C. taurus fluctuated each year, with 2022/2023 having higher numbers on mean and 2021/2022 having lower numbers on mean (Table 3). The mean MaxN of C. taurus was constantly higher at RR than at QM. This difference was especially notable in December 2021, with a mean of 18 individuals per day at RR compared to 9 at QM, and in January 2022, when RR had a mean of 33 individuals per day vs only 1 at QM. Regarding shark abundance, RR had the highest numbers, with 114 individuals of C. taurus observed in a single day, followed by QM (n = 55) and lastly MR (n = 14) (Table 3). Significant differences in MaxN values were observed among the reef sites, according to the Kruskal-Wallis test (H = 61.52, p < 0.0001). The mean MaxN per day showcases some monthly fluctuations, but with consistent peaks observed during the summer months, highlighting a recurring trend in shark abundance. December, January, and February consistently emerge as months with higher MaxN values across sampling seasons (Fig. 7).

Table 3 Variation in Carcharias taurus’ density at different reefs over five sampling seasons.

The table displays the highest maximum number (highest MaxN) and the average maximum number (mean MaxN) per day recorded at Quarter-Mile Reef (QM), Raggie Reef (RR) and Mushroom Rocks (MR) across five consecutive sampling seasons. Instead of listing all individual MaxN values, only the highest value recorded per season is presented, indicating the peak shark abundance at each site.

Sampling season	QM	RR	MR	
Highest MaXN	Mean MaxN	Highest MaXN	Mean MaxN	Highest MaXN	Mean MaxN	
18/19	13	4.9	–	–	–	–	
19/20	48	8.5	41	24.6	11	6.4	
20/21	44	4.2	45	17.8	–	–	
21/22	48	2.6	62	11.2	14	6.1	
22/23	55	10.2	114	23.1	–	–	

Figure 7 Seasonal variation in Carcharias taurus abundance at three sites (QM, RR and MR).

Mean MaxN per day across sampling months, over five consecutive years (2018 to 2023). Data is shown for study sites Quarter-Mile Reef (QM), Raggie Reef (RR) and Mushroom Rocks (MR).

Discussion

Otway et al. (2003) defined aggregation sites for C. taurus as places where five or more individuals were consistently observed across years. However, this definition lacked consideration of seasonal aggregations. In this study the results indicate that the three surveyed reefs in the IWP meet the criteria for seasonal C. taurus aggregation sites, as proposed by Hoschke & Whisson (2016). Aggregation behavior is common among elasmobranchs, and this species is no exception, being well-known for its gregarious nature (Olbers & Cliff, 2017; Haulsee et al., 2016), including pregnant females, based on observations in other various parts of the world (Hoschke & Whisson, 2016; Bansemer & Bennett, 2009). Reducing predation risk is another potential benefit of such aggregations (Guttridge et al., 2012). Considering the large size of these sharks (±190 cm PCL) and the scarcity of natural predators, it seems improbable that they congregate to reduce risk of predation. Foraging efficiency is frequently improved by group behavior (Dewar et al., 2008) yet the presence of hydroid growth on the teeth of these sharks suggests that they may not feed extensively during these aggregations (Pollard, Lincoln-Smith & Smith, 1996), relying on energy reserves stored in their large livers to nourish the embryos (G. Cliff, 2024, personal communications).

Population assessment

In this study, 574 distinct individuals were identified from just three sites in the IWP, over a 5-year sampling period, with no sign of the discovery curve attaining an asymptote. Our study showed 569 mature females, which contrasts with research from the east coast of Australia, where only 271 mature females were identified across 19 aggregation sites (Bansemer & Bennett, 2011).

Throughout the 5 years of survey, the highest numbers of identified individuals and the highest average MaxN per day were recorded in the summer months of December, January and February. Local water temperatures are highest during this period, peaking in February (Staiger, 2020), which enhances embryo development and reduces gestation time, especially in ectothermic species (Bass, 1975; Bansemer & Bennett, 2009). This trend is evident in other coastal shark species, where pregnant females also aggregate in warm waters to accelerate gestation, displaying seasonal fidelity to such sites, such as the leopard shark, Triakis semifasciata, in southern Carolina, USA (Nosal et al., 2014) and the tiger shark, Galeocerdo cuvier in the Bahamas (Sulikowski et al., 2016).

Reproductive cycle

The population of C. taurus in the IWP between September and March is predominantly female (n = 99%). This aligns with the findings of Dicken, Smale & Booth (2006a) who observed that C. taurus catches by anglers from Sodwana Bay to Richards Bay between November and February, were dominated by large females. The clear dominance of females is because most of the mating takes place to the south of the IWP and only the females continue northwards, while the whereabouts and movement patterns of mature males outside of the mating season remains uncertain (Dicken, Smale & Booth, 2006a). Over the course of 5 years, 97% females appeared to be pregnant. Those not visibly pregnant at the time were probably in the early stages of pregnancy, where signs were not yet visible. Non-pregnant females in the resting phase of their reproductive cycle are unlikely to occur in IWP, based on the findings of Dicken, Smale & Booth (2006a), who reported ahigh incidence of adult females in summer in the Eastern Cape.

Research has demonstrated a biennial reproductive cycle for this species in both the NW Atlantic (Branstetter & Musick, 1994; Henningsen et al., 2004), SW Atlantic (Lucifora, Menni & Escalante, 2002); and Australia (Hoschke & Whisson, 2016). The present study adds an additional perspective, as there is evidence of both biennial and triennial reproductive cycles. Three pregnant females were resighted after a 1-year interval, which is in accordance with the well-documented biennial cycle, involving a 1-year resting period. Yet, nine females were resighted only after a 2-year interval, which is suggestive of a triennial cycle, with a 2-year resting period. This is not surprising, as the triennial cycle was also described for nine females in the Australian east coast (Bansemer & Bennett, 2009). These findings have important repercussions for a species which, based on a 2-year cycle, already has a low reproductive output but now it may be further reduced by a one third. This could result in reduced population recruitment and increased vulnerability to threats, such as habitat loss and overfishing. As shown in other studies, elasmobranchs are particularly vulnerable to overfishing and possess a highly limited capacity for population recovery, compared to most teleosts (Myers & Worm, 2005; Stevens et al., 2000). Future research should focus on explaining the drivers of this variability in such cycles and its implications for the population. Additionally, it would be valuable to investigate whether the triennial cycle is associated with smaller females, as our study did not assess female size. Smaller individuals might potentially outgrow this pattern and revert to a biennial cycle with age.

Description of movements

Throughout the study period, the observed movements almost exclusively involved mature female C. taurus presumed to be in a pregnant state. Since there are movements among sites, there is no site fidelity to a specific reef. This is in contrast to the findings of Klein et al. (2019) who demonstrated the existence of genetically differentiated nursery areas, with female C. taurus returning to specific sites, thereby exhibiting reproductive philopatry. To date, no studies have explored whether pregnant C. taurus consistently travels with the same group of individuals.

Most of the northward movements were between December and January, while southward movements were predominantly observed between February and March. These migration patterns align with literature documenting the coastwise movements of pregnant females in South Africa (Bass, 1975; Dicken et al., 2007) and in the southwest Atlantic, where the pregnant individuals are found aggregating in subtropical waters (Sadowsky, 1970), and after parturition, they move south where they rest, in cool waters (Lucifora, Menni & Escalante, 2002).

On a broader scale, knowledge of coastal migration patterns is essential for efficient management (Bonfil, 1997), but unfortunately, available data is frequently lacking. For coastal sharks that migrate seasonally, like C. taurus, this problem is particularly important (Speed et al., 2010). Once again, this study has highlighted the value of Marine Protected Areas (MPAs) in protecting coastal sharks, especially for species that rely on specific areas for reproduction or feeding (Speed et al., 2010). Pregnant C. taurus spend much of their gestation in South Africa’s largest coastal MPA, the iSimangaliso Wetland Park.

Spatial distribution

Raggie Reef had higher sighting rates than the other two sites throughout the study period, with a remarkable 90% presence of C. taurus. These results can be attributed to its geographical location (Fig. 1) and size. Being the southernmost and by far the largest of the three study sites, it is possibly the first reef in the IWP to receive pregnant females coming from the south, and also the last known stop before heading back south to pup. Most importantly though, it is zoned as a sanctuary area, i.e., no fishing, or diving activities are permitted in the IOWZ (Republic of South Africa, 2019), with the exception of bona fide research, as was the case for this project. This sanctuary status ensures that pregnant females are very rarely disturbed at RR. On the other hand, diver disturbance is a problem at Quarter Mile Reef, which has long been recognized as an important site for C. taurus. Because of its extremely close proximity from the launch site, scuba diving with C. taurus at QM has become a popular dive attraction in Sodwana Bay (Dicken, 2014). This proximity to the launch site is a source of disturbance, (Koper & Plön, 2012). The small vessels used at Sodwana Bay are all equipped with twin outboard engines and are required to negotiate the surf zone at high speed, generating noise that falls within the peak sensitivity range of sharks (Casper, 2006). These two factors combined could adversely influence the numbers of sharks at QM. This resulted in the introduction of a park management initiative to close the reef when the sharks first arrive at QM and to only open the reef to scuba diving when the sharks appear to have settled (Olbers & Cliff, 2017).

The three aggregating sites are the only ones known in an MPA which spans some 180 km of coastline. It is also uncertain why these reefs are preferred over others, despite extensive surveying of other reefs. One new site was discovered in January 2022, where an aggregation of close to 50 individuals was found at a new location, aptly named Raggie Garden, a short distance offshore from RR. This reef is at 22 m, far deeper than the three sites used in this study, but well within the depth range (<40 m) of this species (Bennett & Bansemer, 2004; Otway & Ellis, 2011; Hoschke & Whisson, 2016). Although the newly discovered reef was not included in this study, the search for more aggregation sites is ongoing. It is conceivable that the sharks may be forced into deeper water by large swells associated with adverse weather conditions, which may include the effects of cyclones in the summer months, hence sites such as Raggie Gardens become very important at such times.

Conservation

Since our discovery curve has not reached an asymptote after 574 identifications, it supports the conclusion that the South African population seems to be the last remaining stable subpopulation of C. taurus globally, with a genetics-based study indicating that the species has shown no decline in the region (Klein et al., 2020). By contrast, Otway, Bradshaw & Harcourt (2004) indicated that Australia’s south-eastern coast population has experienced severe declines (minimum of 300, most likely <1,000 individuals), with quasi-extinction predicted to be less than 45 years away, unless all anthropogenic, largely fishing-related, mortalities are eliminated. Currently, recreational shore angling for this species is permitted in the IWP, with the stipulation that all the catches are returned to the water alive, but the possibility of post-release mortality cannot be eliminated (Dicken, Smale & Booth, 2006b; Otway & Burke, 2004; Bansemer & Bennett, 2010). Therefore, angler education is important to ensure correct handling. Even if survival rates of released sharks are high, females may still lose their embryos through abortion (Adams et al., 2018).

Other conservation initiatives include an improved code of conduct for divers to minimize disturbance or stress, which may cause sharks to relocate from aggregation sites like QM or MR (Olbers & Cliff, 2017). Other studies have shown that scuba diving affects the behavior and distribution of C. taurus (Barker & Williamson, 2010; Barker, Peddemors & Williamson, 2011). In this regard, smaller groups of recreational divers, governed by a clear code of conduct around the sharks, would be highly beneficial.

Conclusions

This 5-year study has resulted in the compilation of an extensive database to monitor the use by pregnant C. taurus of the iSimangaliso Wetland Park, where they spend much of their gestation. It has contributed to a better understanding of localized movements and the distribution of pregnant females at only three known aggregation sites within this MPA, which spans 180 km of coastline. Our findings underscore the reproductive importance of this region of the South African coastline, providing a foundation for informed conservation strategies that must ensure that the South African population of C. taurus remains healthy, in sharp contrast to the Critically Endangered status of others. Further studies should expand sampling across known sites to reduce data gaps and improve understanding of movements between sites. Ongoing monitoring will continue, thereby providing an estimate of the size of the mature female population and to assess the relative importance of the biennial and triennial reproductive cycles. Additionally, locating and monitoring more sites is crucial for understanding fine-scale habitat use within the MPA and the factors influencing movement between sites.

Supplemental Information

Supplemental Information 1 Database of Carcharias taurus’ individuals in iSimangaliso Wetland Park (2018–2023): sex, sightings, and pregnancy status.

Raw data on all shark individuals identified during the 5-year study period (2018–2023): details on each individual’s sex (female or male), whether they were observed in more than one season (yes or no), the number of sightings per season, the total number of encounters, and, for females, their pregnancy status.

We commend the Sharklife interns for their unwavering support and for diligently collecting data.

Additional Information and Declarations

Competing Interests

Author Contributions

Field Study Permissions

Data Availability

Jennifer M. Olbers is employed by WILDTRUST and an Honorary Research Associate with Nelson Mandela University. Geremy Cliff contracts to WILDTRUST and is also an Honorary Research Associate of the University of KwaZulu-Natal. Grant Smith is employed by Sharklife Conservation Group (managing director). Michelle Carpenter was consulting to Sharklife Conservation Group at the time of the study.

Sara C. Cerqueira performed the experiments, analyzed the data, prepared figures and/or tables, authored or reviewed drafts of the article, and approved the final draft.

Jennifer Margaret Olbers conceived and designed the experiments, authored or reviewed drafts of the article, and approved the final draft.

Grant Smith conceived and designed the experiments, performed the experiments, authored or reviewed drafts of the article, and approved the final draft.

Michelle Carpenter performed the experiments, authored or reviewed drafts of the article, and approved the final draft.

Mário J. Pereira analyzed the data, prepared figures and/or tables, authored or reviewed drafts of the article, and approved the final draft.

Geremy Cliff analyzed the data, authored or reviewed drafts of the article, and approved the final draft.

The following information was supplied relating to field study approvals (i.e., approving body and any reference numbers):

The iSimangaliso Wetland Park Authority and Ezemvelo KZN Wildlife approved the field work with an authorized Research Agreement (Permit RES2023/56).

The following information was supplied regarding data availability:

The raw data is available in Supplemental File.

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
