# Peer review of "The movement and distribution of pregnant spotted ragged-tooth sharks, Carcharias taurus, in the iSimangaliso Wetland Park, South Africa"

_PeerJ, doi:10.7717/peerj.18736_

## Round 0.1 · original submission · Major Revisions

Dear Dr. Cerqueira

You can find the comments and suggestions of the expert reviewers in the attached reports. As you will see, expert reviewers have pointed out the critical errors; therefore, a major revision is needed for your article.

I request you check and correct the manuscript according to the reviewers' reports.

Sincerely

Reviewer 1 ·

Basic reporting

The manuscript was drafted with clear and professional english.

Present referencing is correct, and two more references added.

All figures and tables are relevant and necessary.

The hypotheses and results are well matched.

Experimental design

Experimental design is proper, however, the authors needed to clarify why they selected to take photos from a single-side, or they need to give a reference for single-side imaging.

Validity of the findings

The present findings provide a novel insight to the reproduction of Carcharias taurus, its' pregnancy in particular.

Conclusions are well reported and properly linked to the original research.

Additional comments

Correction notes are added to pdf as popup notes.

Annotated reviews are not available for download in order to protect the identity of reviewers who chose to remain anonymous.

·

Basic reporting

Dear editor,
I have carefully reviewed article suggestion #102837. I think it is a very revealing study, focusing on a critically endangered species and its habitat, if the recommendations are taken into consideration by the authors. Therefore, it is unacceptable for publication in its current form. The article proposal needs many "minor" and few "major" corrections.
Corrections and suggestions in the text are indicated on the Word file itself (peerj-102837-Manuscript_Cerqueira_et_al), using the "track changes" command. In addition, suggestions for figures and tables are shown on a word (peerj-102837-Table_1) and a pdf format file (peerj-reviewing-102837-figures and tables). Later, these three files were combined into a single pdf file format (peerj-102837-referee.pdf) and uploaded to the system to be presented to the attention of the authors.
Minor(s):
- The language of the article needs to be improved. The sentences should be made more clear. Some sentences have been corrected in the text for illustrative purposes.
- Literature that differs in its citation style in the text and in the reference list should be checked. Literature not cited in the text should be removed.
- Suggested corrections to figures and tables should be taken into account.
- The information marked in the "Introduction" and "M&M" sections should be moved to the appropriate places in the "Discussion" section.


Major(s)
- PLEASE EXPLAIN! What standards were adopted as criteria when using the UVC technique? Were the observation hours the same and carried out in a standard manner (for example, between 09:00 and 12:00 in the morning)? Were transects/transects determined for the observations (for example, 100 m long) and were the observations/counts always made on the same line?
- PLEASE CHECK AND EXPLAIN! The text says "the lowest score representing the closest match to the unknown individual" (lines 156-157). The next section explains the high scores (above 20-30). At this point, confusion arises: "scores below 10 were almost always accurate, with a score of 5 being highly likely to indicate the same individual". Now, what should be explained by the "lowest score"? If a score of "5" represents the "same individual", what does "the lowest score represented the closest match to the unknown individual" mean?
- PLEASE CHECK AND EXPLAIN! The section under the heading "Sampling effort" and the data presented in Table 1 are quite confusing. What should the reader understand by the term "Sampling effort" here? Are we considering the number of sightings as 1200 (data in lines 28 and 189)? Or the number of observations as 898 (total in Table 1 as 359+484+55)? Or the number of identified sharks as 574? Or the number of sampling days as 594 (482+76+36 days)?
- In addition, in order to make Table 1 more explanatory and to enable a more accurate comparison of the study data, the following is suggested. Researchers should standardize their data as "number of sharks" per "observation hour" or "number of observation days". Thus, the actual effort per unit effort can be seen (as CPUE). A CPUE pre-calculation was made by me by adding to Table 1 (please see additional word file; peerj-102837-Table_1). In these calculations, the highest effort was in RR.. Researchers should use such an approach.
- Please review the additional file (peerj-102837-Table_1) for table editing and suggestions.
- If the authors decided to use a CPUE, they should explain it in M&M section.

Experimental design

PLEASE EXPLAIN! What standards were adopted as criteria when using the UVC technique? Were the observation hours the same and carried out in a standard manner (for example, between 09:00 and 12:00 in the morning)? Were transects/transects determined for the observations (for example, 100 m long) and were the observations/counts always made on the same line?
PLEASE CHECK AND EXPLAIN! The section under the heading "Sampling effort" and the data presented in Table 1 are quite confusing. What should the reader understand by the term "Sampling effort" here? Are we considering the number of sightings as 1200 (data in lines 28 and 189)? Or the number of observations as 898 (total in Table 1 as 359+484+55)? Or the number of identified sharks as 574? Or the number of sampling days as 594 (482+76+36 days)?
n addition, in order to make Table 1 more explanatory and to enable a more accurate comparison of the study data, the following is suggested. Researchers should standardize their data as "number of sharks" per "observation hour" or "number of observation days". Thus, the actual effort per unit effort can be seen (as CPUE).

Validity of the findings

PLEASE CHECK AND EXPLAIN! The text says "the lowest score representing the closest match to the unknown individual" (lines 156-157). The next section explains the high scores (above 20-30). At this point, confusion arises: "scores below 10 were almost always accurate, with a score of 5 being highly likely to indicate the same individual". Now, what should be explained by the "lowest score"? If a score of "5" represents the "same individual", what does "the lowest score represented the closest match to the unknown individual" mean?
- PLEASE CHECK AND EXPLAIN! The section under the heading "Sampling effort" and the data presented in Table 1 are quite confusing. What should the reader understand by the term "Sampling effort" here? Are we considering the number of sightings as 1200 (data in lines 28 and 189)? Or the number of observations as 898 (total in Table 1 as 359+484+55)? Or the number of identified sharks as 574? Or the number of sampling days as 594 (482+76+36 days)?

Additional comments

Please review the additional file (peerj-102837-referee.pdf)

Reviewer 3 ·

Basic reporting

This manuscript describes patterns of presence of Carcharias taurus at several aggregation sites in Eastern South Africa based on photo ID of individuals. This is an interesting study and does advance understanding of migratory patterns of these sharks, particularly for mature females and the reproductive cycle of this species. The findings are interesting and a convincing confirmation of what was suspected about movements of these sharks.

Experimental design

What is not as convincing as presented in this version of the manuscript is the certainty with which individuals were positively identified since there is no mention of blind tests to quantify ability to correctly identify individuals or additional descriptions of success of the photo IDs. As written, the authors would have the reader believe that they were successful in identifying every individual examined. If that’s the case then that’s impressive, especially given the length of time that the study spanned. That means that the authors are confident that ontogenetic changes in individuals (and their spots) never obscured the ability to identify an individual over half a decade. A second feature of the paper that is not as obvious to a reader as it may be to the authors, is the designation of pregnant females as opposed to non-pregnant females. In real life observation of the individuals it may be very clear whether a shark is pregnant or not, but there is no description of determination of pregnancy in the methods and the main criteria seems to be the presence of mating scars or a distended mid-region. If it is that obvious that a shark is pregnant, then the authors could make that point more clearly.

Validity of the findings

The last major comment has to do with what seems to be a struggle on the part of the authors to indicate what the importance of the study is. There are comments about migrations, movement patterns and even philopatry, but the study actually provides little information on these topics because it is limited to one demographic within a small area of their range. There is very little actual data on movements of individuals. The authors also provide estimates of abundance, presenting data on the number of individuals at different times, in an attempt to link these data to the size of the population. That is obviously quite a stretch from this small area and pregnant females to the entire population. There is also quite a bit of discussion about conservation from the point of view of changes that could be established as well as future studies. Much of this discussion is extraneous to the real importance of this study, comes across as text that someone thought would be good to include to make the project seem important and obscures what this study accomplishes and how it advances understanding of these sharks. This study demonstrates the importance of these areas for reproduction of a species that is vulnerable to population decline.

Additional comments

If the methods are described more clearly, with more detail, especially with an intent to convince the reader of their validity, that would be an improvement.
If the text in the discussion was limited to topics that data in this study provide evidence for or against, the text could be reduced, the main findings of the study could be more focused and extraneous comments that are largely speculation or that are not really supported by the study could be reduced.

---

## Round 0.2 · accepted · Accept

Dear Dr. Cerqueira

I thank you for making the corrections and changes requested by the reviewers. I read and checked your valuable article carefully and am happy to inform you that the article has been accepted for publication in PeerJ.

(By the way, when the proofing PDF is received, finally check and correct the Ebert et al (2022) in the references section.)

Sincerely yours

Reviewer 1 ·

Basic reporting

The authors completed the revision in a proper manner, and some of their rejections to some previous correction comments are acceptable.
However, in lines 461 and 462 the reference of Ebert et al. (2021) requires correction as follows:

Ebert, D. A., Dando, M. & Fowler, S. (2021). Sharks of the world, a complete guide. Princeton, Wild Nature Press

The remaining terms and conditions seen on the right of this message panel are fullfilled properly through the text.

Experimental design

With the addition of Rezzolla et al. (2014) reference, now the UVC methodology has a solid scientific background.

The remaining terms and conditions seen on the right of this message panel are fullfilled properly through the text.

Validity of the findings

The remaining terms and conditions seen on the right of this message panel are fullfilled properly through the text.

Additional comments

None

Annotated reviews are not available for download in order to protect the identity of reviewers who chose to remain anonymous.

·

Basic reporting

I have carefully read the MS. Thanks to the authors for considering the suggestions and providing more clear information. I found this version more unambiguous, and professional English was used throughout. Literature references and sufficient field background/context provided. Professional article structure, figures, tables. Raw data shared. Self-contained with relevant results to hypotheses.

Experimental design

The MS has original primary research within the journal's Aims and Scope. Its research question is well-defined, relevant, and meaningful. The paper states how the research filled an identified knowledge gap.
Methods described with sufficient detail & information to replicate.

Validity of the findings

All underlying data have been provided; they are robust, statistically sound, & controlled.
Conclusions are well stated, linked to the original research question & limited to supporting results.

Additional comments

I have no additional comment.